# Power analysis for personal light exposure measurements and interventions

**Johannes Zauner** [1,2]*, **Ljiljana Udovicic** [3], **Manuel Spitschan** [1,2,4,5]

**1** Department Health and Sports Sciences, Technical University of Munich, TUM School of Medicine and Health, Chronobiology & Health, Munich, Germany, **2** Max Planck Institute for Biological Cybernetics, Max Planck Research Group Translational Sensory & Circadian Neuroscience, Tübingen, Germany, **3** Federal Institute for Occupational Safety and Health (BAuA), Dortmund, Germany, **4** TUM Institute for Advanced Study (TUM-IAS), Technical University of Munich, Garching, Germany, **5** TUMCREATE Ltd., Singapore, Singapore

\* johannes.zauner@tum.de

## Abstract

### Background

Light exposure regulates the human circadian system and more widely affects health, well-being, and performance. With the rise in field studies on light exposure's effects, the amount of data collected through wearable loggers and dosimeters has also grown. These data are more complex than stationary laboratory measurements. Determining sample sizes in field studies is challenging, as the literature shows a wide range of sample sizes (between 2 and 1,887 from a recent review of the field and approaching $10^5$ participants in first studies using large-scale 'biobank' databases). Current decisions on sample size for light exposure data collection lack a specific basis rooted in power analysis. Therefore, there is a need for clear guidance on selecting sample sizes.

### Methods

Here, we introduce a novel procedure based on hierarchical bootstrapping for calculating statistical power and required sample size for wearable light and optical radiation logging data and derived summary metrics, taking into account the hierarchical data structure (mixed-effects model) through stepwise resampling. Alongside this method, we publish a dataset that serves as one possible basis to perform these calculations: one week of continuous data in winter and summer, respectively, for 13 early-day shift-work participants (collected in Dortmund, Germany; lat. 51.514˚ N, lon. 7.468˚ E).

### Results

Applying our method on the dataset for twelve different summary metrics (luminous exposure, geometric mean, and standard deviation, timing/time above/below threshold, mean/midpoint of darkest/brightest hours, intradaily variability) with a target comparison across winter and summer, reveals required sample sizes ranging from as few as 3 to more than 50. About half of the metrics–those that focus on the bright time of day–showed sufficient power already with the smallest sample. In contrast, metrics centered around the dark time of the day and daily patterns required higher sample sizes: mean timing of light below mel

**Data Availability Statement:** All relevant data for this study are publicly available from the Zenodo repository (https://doi.org/10.5281/zenodo.14035242).

**Funding:** The project (22NRM05 MeLiDos) has received funding from the European Partnership on Metrology, co-financed by the European Union's Horizon Europe Research and Innovation Programme and by the Participating States. All authors are part of the project. The project directly funds the position of J.Z. The funders had no role in study design, data collection and analysis, decision to publish, or preparation of the manuscript.

EDI of 10 lux (5), intradaily variability (17), mean of darkest 5 hours (24), and mean timing of light above mel EDI of 250 lux (45). The geometric standard deviation and the midpoint of the darkest 5 hours lacked sufficient power within the tested sample size.

## Conclusions

Our novel method provides an effective technique for estimating sample size in light exposure studies. It is specific to the used light exposure or dosimetry metric and the effect size inherent in the light exposure data at the basis of the bootstrap. Notably, the method goes beyond typical implementations of bootstrapping to appropriately address the structure of the data. It can be applied to other datasets, enabling comparisons across scenarios beyond seasonal differences and activity patterns. With an ever-growing pool of data from the emerging literature, the utility of this method will increase and provide a solid statistical basis for the selection of sample sizes.

## Introduction

Exposure to light has profound influences on human physiology and behavior. It is critical in regulating the circadian system, affecting health, well-being, and performance [1]. Much of our current understanding of the effects of light on humans is based on highly controlled laboratory studies [2]. However, these effects are modulated by various factors, including the prior light exposure received before the laboratory, also called prior light history [3–6]. This makes it challenging to gauge real-world impacts from findings and to replicate all relevant factors in laboratory studies [7]. Field studies and observational studies in real-world contexts provide valuable insights into the complex environmental and behavioral patterns that determine the everyday light and optical radiation exposure of humans and can link those patterns to relevant outcomes [8]. However, stationary laboratory measurements do not fully capture the variability of light exposure that individuals experience in real-world settings.

Wearable light exposure loggers and optical radiation dosimeters come closer to approximating the objective measure of interest, namely retinal exposure. These devices have been around for several decades, but early studies using them have been few and limited to small sample sizes [9]. Developments in the field of miniaturized sensor platforms have reduced the cost and size of these devices while computational power and interfaces improved. We have seen a surge in studies using wearable light sensors in recent years, likely fueled by these developments combined with increasing scientific community interest [10–18]. This trend is expected to continue, introducing new challenges for researchers studying the non-visual effects of light. Examples include studies drawing from large-scale data sources of tens of thousands of people from research biobanks, linking increased light exposure at night with increased risk for metabolic issues (e.g., type 2 diabetes in Windred et al. [18]) or several mental pathologies (e.g., major depression, anxiety disorder, bipolar disorder, and others in Burns et al. [13]). In contrast, increased light exposure during the day reduced the risk significantly for many of these aspects. Still, smaller studies with less than 100 participants, such as from Didikoglu et al. [12], are valuable, as they are designed specifically for light exposure, showing that participants are regularly outside recommended exposure values during the day, evening, and nighttime [1] and how light exposure is related to other behavioral patterns such as sleep [12].

As the field moves from single time-point "spot" measurements of light to time series of light measurements, new analysis tools and methods are being introduced. New metrics and analytic strategies have been proposed, but a coherent framework is missing [19]. Notably, a proper understanding of how human physiology integrates light exposure patterns in the real world over time requires time-series data. These data are considerably "messier" as the collected data will routinely contain valid and non-valid timeframes, such as the non-wear times of the device. Since light exposure varies across time within each participant, the data structure is inherently hierarchical. All these topics not only affect the analysis of light exposure data but also impact the study design.

A fundamental open question for studies using light exposure measurements or dosimetry data is how many participants are required when measuring personal light exposure. This question is not limited to, but is especially important for, experiments with interventions targeted to change light or optical radiation exposure. In field experiments, researchers can seldom control exposure directly but can influence factors affecting exposure [20], e.g., changing artificial or daylight settings or behavioral patterns [21–23].

This article presents a novel approach for estimating statistical power grounded in analyzing historical data through bootstrapping. Although not a new method [24–26], bootstrapping is rarely used in this field, requiring both suitable historical data and appropriate implementation, which are currently lacking. Currently, sample size determination in studies using wearable light exposure loggers is often unrelated to statistical power [10,13,17], not stated at all [11,15,16], or is based on general recommendations that take outcomes from unrelated fields, metrics, and effect size [12]. This method extends the literature on considerations for power analysis specific to non-visual effects of light, similar, e.g., to the work of Spitschan et al. [27] on power analysis for human melatonin suppression experiments.

General recommendations for selecting sample sizes outside the light logging or dosimetry field have been developed Maas and Hox [28]. This article introduces a new approach to estimating statistical power based on historical data analysis using bootstrapping. Those recommendations are hardly applicable in all cases, however. The power of a study is partly determined by how sensitive a specific metric is to measure change in a research scenario relative to the intra- and inter-individual variance of the metric (which, in the context of hierarchical or multilevel analysis, is expressed as lower- and higher-level variance, respectively).

Kumle et al. [26] provide an open introduction to power calculations for hierarchical data with different methods, some of which rely on a robust statistical model built on historical data, and others are based on theoretical considerations of relevant effect size.

For light or optical radiation exposure data, the choice of metric depends on the research question and the spectral quantity of interest. The raw time series of measurements from wearable devices is hardly ever the primary measure of a study. Instead, it is the basis for a metric calculated from it, capturing a construct of interest. For example, light exposure data might be used to calculate the cumulative light dose (luminous exposure), the mean timing of light exposure, or the time above a threshold, such as 250 lx or 1000 lx melanopic equivalent daylight illuminance (mel EDI [29]). These have different theoretical motivations, such as capturing the stability of light exposure patterns, the dynamic range between day and night, or light exposure in its stricter sense, i.e., the dose.

A systematic review by Hartmeyer and Andersen [19] recently surveyed the landscape of light exposure studies and found over forty different metrics in use, some of which were further parametrized, yielding more sub-metrics. Due to this "metaverse" of light exposure metrics, it is difficult to compare studies and estimate the sample size required to detect the effect of light on the circadian system based on previous publications.

To develop a framework for statistical power estimation in this field, we will use partial data from a previously published study employing personal light exposure data [16] and provide a suitable implementation for power analysis. A natural experiment, i.e., the change in light exposure between seasons, will be a proxy for a hypothetical research interest. We will use the data to estimate the sample size required to detect a change in a light exposure metric between seasons. We will use a variety of metrics calculated from the raw light exposure data and compare the results, thereby demonstrating the impact of the choice of metric on the required sample size. The scope of this method is currently limited to changes in light exposure metrics, not to changes in other outcomes (e.g., sleep quality, alertness) dependent on light exposure metrics. The dataset used in our example analysis consists of 182 person-days for 13 participants across two seasons (collected in Dortmund, Germany, lat. 51.514˚ N, lon. 7.468˚ E) and will be published alongside this paper.

## Materials and methods

### Overview

The proposed method can be used to calculate the necessary sample size to reach the required statistical power level (here: 0.8, 80%) for a range of relevant metrics of personal light exposure. The effect size under evaluation is taken from a historical dataset where participants are compared between the summer and winter seasons. The method uses bootstrapping [25] at its core, i.e., sample reuse techniques, and importantly, takes higher level (inter-individual) and lower level (intra-individual) variance into account. All metrics use melanopic equivalent daylight (D65) illuminance (mel EDI) as their basis for calculation, defined by the CIE S 026 standard [29].

### Software

We used the *R* software [30], Version 4.2.3, for all analyses. Data pre-processing and visualizations were done with the *LightLogR* package [31,32], specifically designed to analyze wearable light exposure logger data. The statistical analysis is based on a linear mixed-effect model, performed with the *lme4* package [33], and *p* values are calculated with the *lmerTest* package [34]. All scripts and results are part of *S1 File* as an HTML Quarto document [35], including the necessary source code for replication. Reproducibility is ensured through CODECHECK [36].

### Dataset

The data used is part of a historical dataset collected for a study evaluating the personal light exposure of shift workers in two geographical locations (London and Dortmund) [16]. Light exposure was recorded using the *Actiwatch Spectrum* device (*Philips*, Amsterdam) attached to the outer layer of clothing at chest level. The data relate to the Dortmund geolocation branch of that study and were collected in a field experiment, where participants recorded personal light exposure continuously for a working week with different shift work settings in Winter (Jan/Feb), Spring (Apr, not part of the dataset), and Summer (Jun) 2015. For a more comprehensive description of the study and participants, we refer to the original publication by Price et al. [16]. Table 1 details the participants chosen for this analysis, described below under *Data pre-processing*. The data used in the present study are published under a Creative Commons license (CC-BY) on Zenodo [37].

**Table 1. Overview of participant data for chosen analysis group.** Subject Group: Nurses (both male and female); all indoor workers (morning shift); various departments (e.g. Dermatology, Cardiac Surgery, Gastroenterology) of the Dortmund Clinic (Dortmund, Germany; 51.514˚N, 7.468˚E); Measurements of light exposure both during and outside of work hours for one week–January and June 2015; NA: Not available.

| Sex (f/m) | Age (yr) | BMI |
| --- | --- | --- |
| f | 37 | 22.0 |
| m | 46 | 26.6 |
| m | 39 | 24.2 |
| f | 27 | 23.1 |
| m | 44 | 23.4 |
| f | 23 | 31.5 |
| f | 23 | 25.8 |
| f | 24 | *NA* |
| f | 42 | 34.7 |
| f | 22 | 23.4 |
| f | 54 | 22.7 |
| f | 46 | 30.8 |
| f | 32 | 20.8 |

## Data pre-processing

**Scope.** Data pre-processing aims to load and clean the data files exported from the measurement device. Melanopic EDI values are then calculated from sensor data to derive relevant metrics [29].

**Pre-selecting participant days.** From the potential pool of participants, 13 participants were pre-selected for the analysis at hand based on the following criteria:

- At least two days of data, during which the participant worked in the early-day shift (i.e., no night or late shifts), during the January and June data collection periods.

An early-day shift (day shift from here on out) is defined as working from 06:00 in the morning until 14:00 local time. See *Fig 1* for an overview. Only workdays were selected for analysis. The rationale for choosing this subgroup is that this early shift resembles non-shift working hours most closely. Including all shift types and further free days would require interaction effects between these parameters. In our particular example, we want to assess the impact of season a specific shift group. This makes the example analysis more straightforward to follow and replicate.

**Non-wear time / non-missing data.** Furthermore, each participant day must have a minimum of 80% valid, non-missing data, or that day will be removed from the analysis. Non-missing data are evaluated by a regular, uninterrupted sequence of measurement intervals (epochs) that are dominant to the data. Valid data are assessed through a dataset column indicating whether the device was worn. If less than two days per season remain, this participant is removed from the analysis. Four days were removed from analysis due to the threshold criterion (see *Fig 2*), leaving the total number of participants at 13 and the number of participant days at 88 (46 in winter, 42 in summer, see *Table 2*).

## Calculating melanopic EDI from sensor data

The Actiwatch Spectrum device has three sensors that record light exposure in the visible spectrum's red, green (G), and blue (B) regions and illuminance. In the study of Price et al. [16] the sensitivities of the G- and B-sensors were linearly combined to yield a spectral match to the

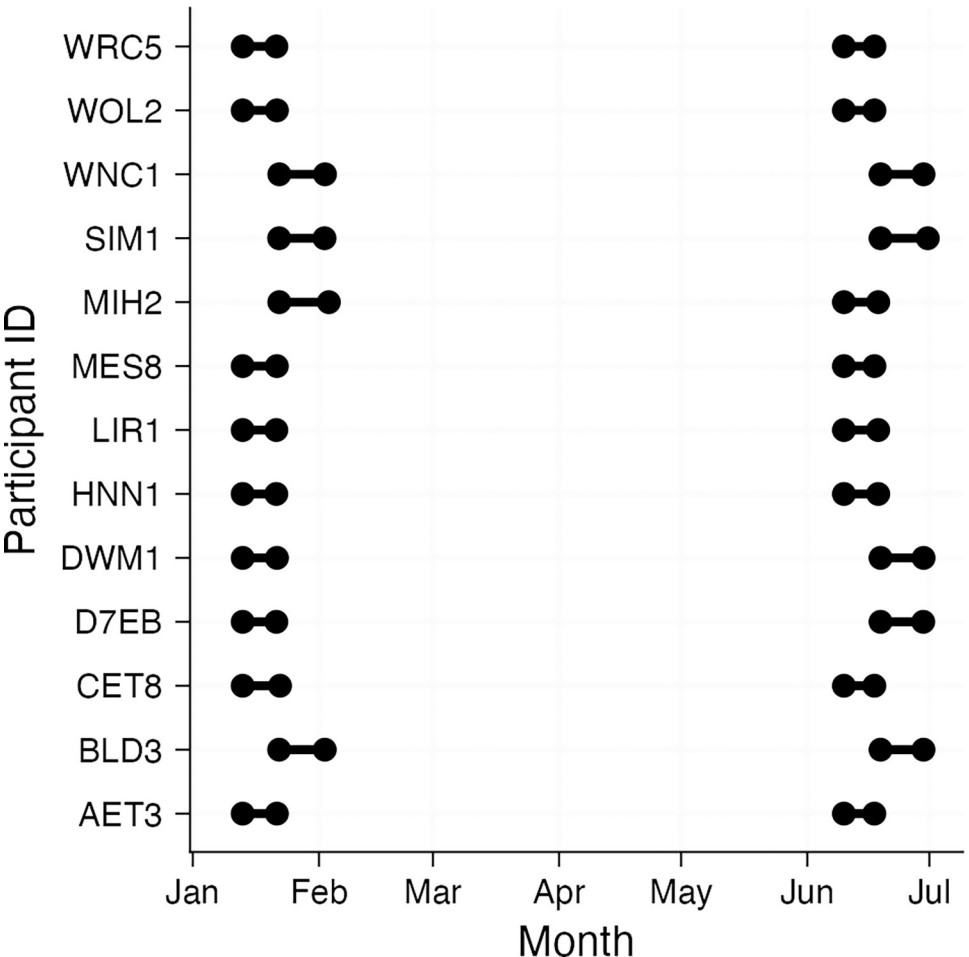

**Fig 1. Overview of available measurement timespans for each participant.** Y-axis: Participants by ID. X-axis: Timeline between January and July. Dots indicate the beginning or end of data for a collection period. Bars indicated times of available data within the collection period.

melanopic action spectrum. Devices were calibrated, and mel EDI was calculated according to:

$$E_{v,mel}^{D65} = \frac{4.3 \cdot (G + B) mW \cdot m^{-2}}{1.3262 \ mW \cdot lm^{-1}} \tag{1}$$

Where G and B are the green and blue sensor outputs, respectively. Lastly, a range of typical wearable light exposure logger-related metrics was calculated for each participant's day [19].

**Table 2. Summary of remaining and invalid days included in the analysis.**

| Characteristic | Valid days, n = 88[1] (96%) | Invalid days, n = 4[1] (4%) |
|---|---|---|
| Season | | |
| Winter | 46 (52%) | 3 (75%) |
| Summer | 42 (48%) | 1 (25%) |
| Invalid/Non-Wear Time | 0%– 15% | 24% - 34% |

[1]n (%); Range.

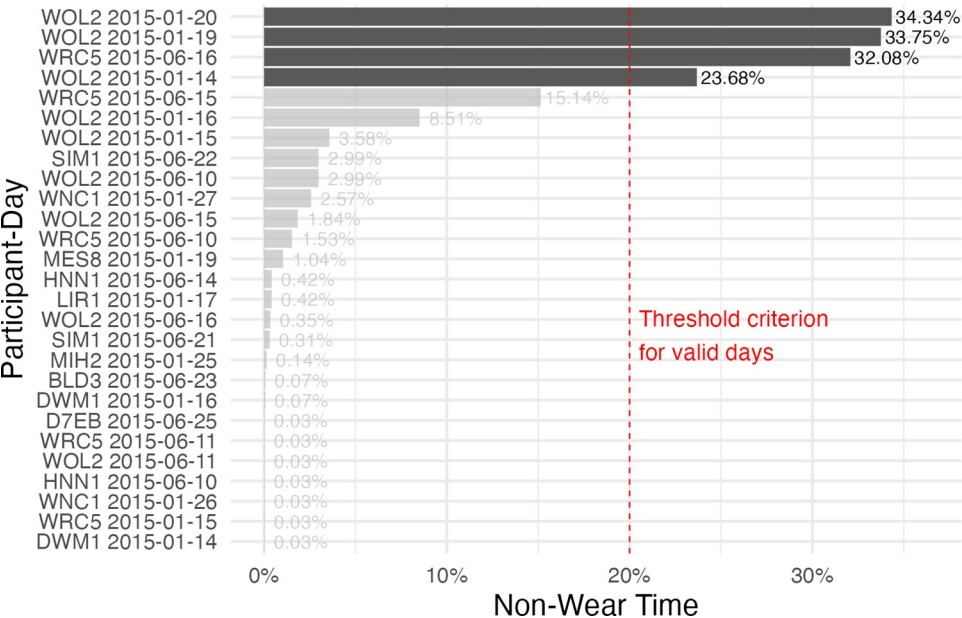

**Fig 2. Summary of invalid / non-wear times across participant days (>0%).** Y-axis: Participant days where the non-wear time is not zero. X-axis: Relative non-wear time per day in percent. The red vertical line indicates the chosen threshold criterion above which a participant day is removed from further analysis. Dark grey bars indicate participant days that crossed the threshold.

These are the basis for the bootstrapping procedure and power analysis. *Table 3* shows an overview of used metrics.

## Bootstrapping

The bootstrapping procedure derives multiple samples *m* for each sample size range *n* by sampling the dataset with replacement, where *n* is the number of participants. This procedure accounts for variance within individuals (intraindividual variance) across time, which differs from traditional bootstrapping procedures, where either no grouping occurs, and common clustered or grouped bootstrapping, where only the lower level is sampled (i.e., grouping by participant and re-sampling within a participant) [25]:

- First, the higher order (participants) is sampled with replacement until *n* is reached

- Second, the lower order (daily metrics within participants and between seasons) is resampled with replacement

This stepwise resampling procedure can also consider more hierarchical levels if needed. The benefit of bootstrapping is the generation of value-pair distributions (winter & summer)

**Table 3. Overview of calculated metrics, their units, and parametrization.**

| Variable name | Unit | Parameters |
|---|---|---|
| Geometric mean & standard deviation (GM, GSD) | lx | log base 10 |
| Luminous exposure (LE) | lx·h | - |
| Time above threshold (TAT) | h | thresholds: 250 lx; 1000 lx |
| Mean timing of light (MTL) | hh:mm | above 250 lx, below 10 lx |
| Intradaily variability (IV) | - | - |
| Conditional mean & midpoint (M10, L5) | lx, hh:mm | brightest 10 hours, darkest 5 hours |

based on real variances captured by the dataset for each metric and depending on **n**. The range of sample sizes **n** in our analysis is 3 to 50, and **m** is 1000.

## Power analysis

The power analysis aims to detect the minimum sample size required to reach 0.8 (80%) power for each metric. Power is calculated as the fraction of significant results compared to all statistical tests at each respective sample size produced by the bootstrapping procedure. The analyzed model uses the equation (mathematical and *Wilkinson* notation, respectively)

$$E(\text{Metric}_i) = \alpha_i + \beta_i * \text{Season}_{\text{Winter}} + b_{p,i}, \qquad \text{Metric}_i \sim 1 + \text{Season} + (1|\text{Participant}) \quad (2)$$

$$\text{where Metric}_i \sim (\mu_i, \sigma^2{}_i), b_{p,i} \sim N(0, \sigma^2_{b,i})$$

Intercept $\boldsymbol{\alpha_i}$ and slope $\boldsymbol{\beta_i}$ are the fixed effects of the model described in Eq 2. The intercept $\boldsymbol{\alpha_i}$ indicates the expected value of a respective metric $\boldsymbol{E(\text{Metric}_i)}$ when all other terms in the equation are zero. The slope, or beta coefficient $\boldsymbol{\beta_i}$, represents the change in a metric's expected value in winter compared to summer. As random effects, we included random intercepts $\boldsymbol{b_{p,i}}$ by participant $\boldsymbol{p}$. Random intercepts $\boldsymbol{b_{p,i}}$ show how much individuals deviate from the average value of a metric ($\boldsymbol{\alpha_i}$) with a *Gaussian* distribution of mean 0 and a variance of $\boldsymbol{\sigma^2_{b,i}}$. We did not include random slopes for computational stability across many tests. Each metric has a *Gaussian* distribution, with variance $\boldsymbol{\sigma^2{}_i}$ around its mean $\boldsymbol{\mu_i}$. The significance level for the fixed effect of *Season* is set at 0.05.

## Results

### Data pre-processing

Fig 3 shows the measurement values from participants with highlighted days that were included and removed from the analysis. Table 4 shows the metrics (median, interquartile range, IQR) based on those days.

### Bootstrapping

576,000 resampled datasets were generated across all metrics and the range of sample sizes (3 to 50).

### Power analysis

Table 5 shows the required sample size to reach at least 80% power within the tested range of sample sizes. Two metrics never reached this level (geometric SD and the midpoint of the darkest 5 hours). Over half of the metrics reached the threshold with the starting sample size of 3, while the rest required sample sizes in between (mean timing of light below 10 lx, intradaily variability, mean of darkest 5 hours, and mean timing of light above 250 lx). Fig 4 shows the distribution of power levels across sample sizes for each metric.

## Discussion

Our results show the large variety of required sample sizes for the same dataset but with different metrics corresponding to different analysis approaches. While we do not recommend small sample sizes, the required sample size varies based on the metric and experimental conditions. A small sample size was sufficient to detect the substantial change in daily light exposure between winter and summer. In most research settings, only a subset of metrics is

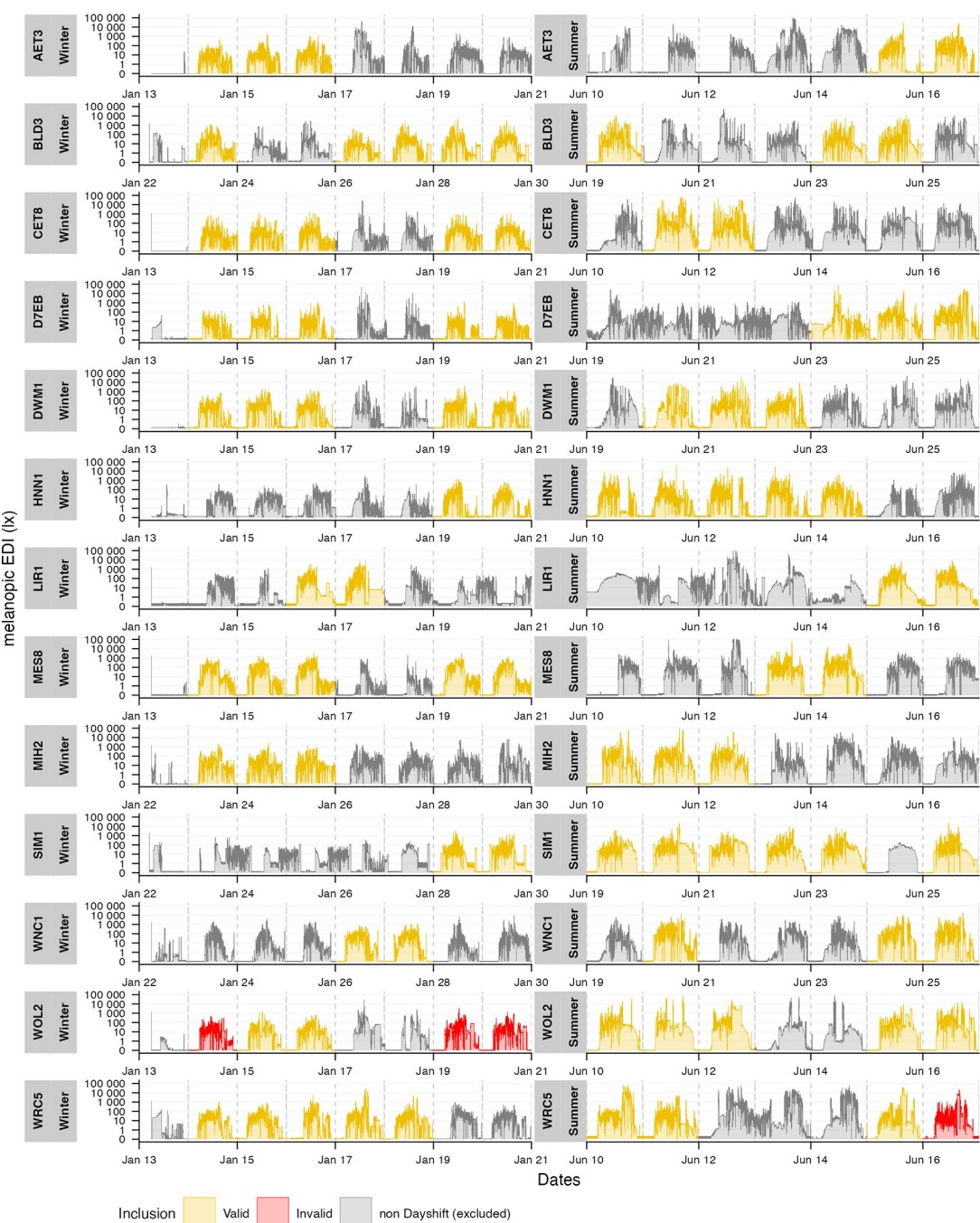

**Fig 3. Measurement values from participants with highlighted days that were included (valid, yellow) and removed from the analysis (invalid, red), as well as non-early-day shift days (excluded, grey).** Facets: Participant data in winter (left) and summer (right). Y-axis: Melanopic EDI in lux. X-axis: Timeline. Dashed vertical lines indicate midnight.

typically necessary. Instead, the method applies to any specific subset of metrics, with fewer metrics allowing for more resamples and greater computational speed.

Other group-level characteristics are also worth considering, such as comparing day- and night-shift instead of season. Some metrics may respond more to seasonal changes, while others may be sensitive to shift schedule variations. The underlying dataset provides the effect size

**Table 4. Metrics for winter and summer seasons.** The table shows median values with their interquartile range. The table also indicates the number of days a respective metric could not be calculated.

| Metric | Winter, N = 46[1] | Summer, N = 42[1] |
|---|---|---|
| Geometric mean (lx) | 1.23 (0.65, 1.94) | 3.39 (2.31, 5.90) |
| Geometric sd (lx) | 26 (14, 135) | 33 (21, 121) |
| Luminous exposure (lx × h) | 705 (462, 1,091) | 4,052 (2,429, 6,881) |
| Time above 250 lx (h) | 0.56 (0.33, 0.90) | 2.34 (1.66, 3.37) |
| Time above 1000 lx (h) | 0.03 (0.00, 0.08) | 0.75 (0.47, 1.21) |
| Mean timing of light above 250 lx (hh:mm) | 13:31 (11:58, 14:20) | 13:37 (12:41, 14:40) |
| Days without metric | 1 | 0 |
| Mean timing of light below 10 lx (hh:mm) | 11:40 (11:08, 12:17) | 12:37 (11:56, 13:10) |
| Intradaily variability | 0.84 (0.63, 1.37) | 1.23 (0.95, 1.60) |
| Mean of darkest 5 hours (L5, lx) | 0.05 (0.01, 0.12) | 0.06 (0.03, 0.14) |
| Midpoint of darkest 5 hours (L5, hh:mm) | 01:57 (01:18, 02:29) | 02:11 (01:28, 02:29) |
| Mean of brightest 10 hours (M10, lx) | 69 (43, 104) | 380 (224, 655) |
| Midpoint of brightest 10 hours (M10, hh:mm) | 11:19 (10:54, 11:44) | 12:52 (11:34, 13:55) |

[1] Median (IQR).

and variance for all hierarchical levels. The bootstrapping procedure maps out the space of credible distributions given the data. Bootstrapping also reveals when hardly an effect is present in the data. This is especially visible in the midpoint of the darkest 5 hours, where not only the threshold power level was not reached. The slope across sample sizes remains flat (see Fig 4J). The underlying differences represent a (close to) null-effect scenario, where a higher sample size is not necessarily better, as the effect might not be statistically or clinically relevant.

Our approach differs from Kumle et al. [26] in several important ways. Firstly, as we lay out the method specifically for light exposure data, the data we provide alongside this publication are already specific to the field, where very few light exposure data are openly accessible at the

**Table 5. Required sample size to reach the threshold power level for each metric.**

| Metric | Required sample size[1] | Power[2] |
|---|---|---|
| Time above 250 lx (h) | 3 | 100% |
| Geometric mean (lx) | 3 | 98% |
| Luminous exposure (lx × h) | 3 | 98% |
| Mean of brightest 10 hours (M10, lx) | 3 | 98% |
| Time above 1000 lx (h) | 3 | 98% |
| Midpoint of brightest 10 hours (M10, hh:mm) | 3 | 81% |
| Mean timing of light below 10 lx (hh:mm) | 5 | 81% |
| Intradaily variability | 17 | 80% |
| Mean of darkest 5 hours (L5, lx) | 24 | 81% |
| Mean timing of light above 250 lx (hh:mm) | 45 | 80% |

*Note*: Did not reach the threshold: **Geometric sd (lx), Midpoint of darkest 5 hours (L5, hh:mm).**

[1] The sample size calculation is based on a bootstrap resampling of daily metrics between winter and summer seasons for 13 participants. For each resampled dataset, significance was tested in a mixed-effect model with a significance level of 0.05. The fraction of significant differences were compared against the power level threshold of 0.8. The required sample size is the minimum sample size that reaches this threshold, with 1,000 resamples per sample size (sample sizes from 3 to 50 were tested). The total amount of resamples/tests is 576,000 across all metrics.

[2] **Power at the required sample size.**

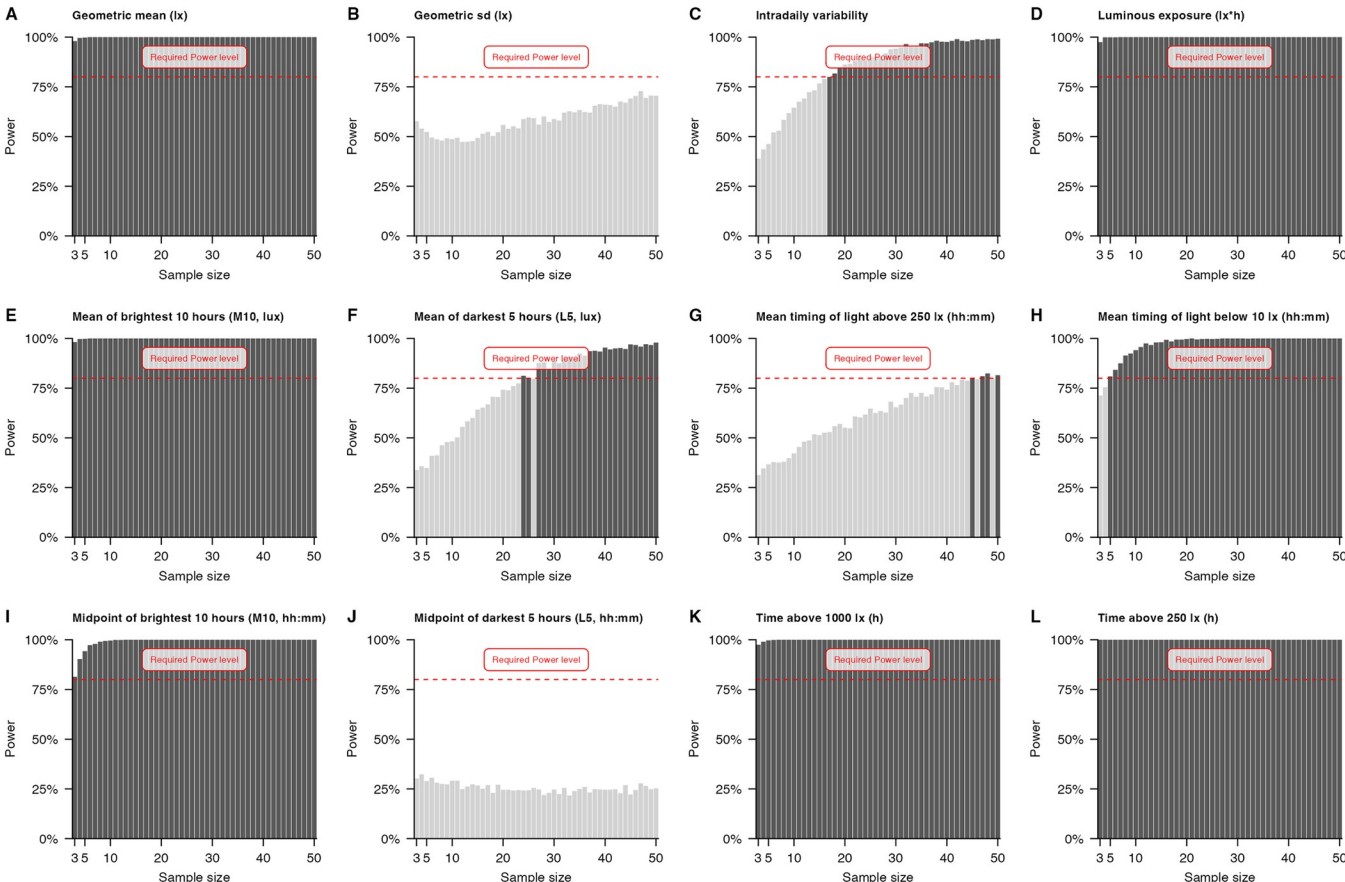

**Fig 4. Distribution of power levels (y-axis) across sample size (x-axis) for each metric in alphabetical order (A-L).** Each panel contains a horizontal dashed red line, indicating the required threshold of 80% power. Light grey bars indicate that this sample size did not reach this threshold; darker grey bars that it did.

time of publication. The main difference between our proposed method and Kumle et al. [26] is our bootstrapping approach to derive a basis for statistical testing compared to their simulation-based approach. The simulation-based approach draws parameters from a (general) mixed effect model (GLMM), meaning that distributional assumptions underlying the (G) LMM affect the power analysis outcome and might not accurately represent the data. Our bootstrapping approach does not require such assumptions, as parameter distributions from the sample are retained with the bootstrap.

Kumle et al. [26] do provide another method that does not rely on (raw) data but on theoretical considerations based on prior studies. However, in the emerging field of personal light and optical radiation exposure research, very few study results can be mapped to one another regarding their reported outcomes. Our approach draws from the underlying exposure data and is thus applicable even in cases of differing metrics.

Currently, sample size determination in research studies collecting wearable light exposure measurements is based on something other than appropriate methods, as was laid out in the introduction. Using bootstrapping for nested data and providing open-access datasets will help establish reliable sample size guidelines for future studies. This method is not limited to light exposure data but can also apply to any application requiring similar analysis of long-term or time-series data (see, e.g., Masuda et al. [38]). The methodological addition to the

bootstrapping procedure is the stepwise resampling from higher to lower levels, where common implementations only resample at one level.

One limitation is the current data set, as it only allows for the singular comparison between seasons. Further, because the work times of the early day-shift workers differ from typical, non-shift work times, the specific results of the analysis in this publication likely do not hold for a population with vastly different work schedules. To generalize, the usefulness of the proposed method heavily depends on the amount of relevant historical data available, which is few for most researchers at present. Light exposure data from various study settings should be made available under open licenses to facilitate further comparisons, also for interventions. If, e.g., a hypothetical lighting intervention is supposed to increase light exposure of office workers in winter to levels like in summer, the described method could be quickly applied.

Another limitation our procedure is that it only applies to changes in light exposure measurement metrics. Many studies connect light exposure (and derived metrics) to other quantities or constructs, such as sleep, alertness, or mental health [10,12,13], and those other quantities will not be available in most datasets. A more theoretical approach to power estimation will be needed, as described Kumle et al. [26]. But even in those cases, historical data can help estimate the (conditional) variance for the light exposure-related metric in question.

A final limitation is that the correct metric cannot be determined through the bootstrapping procedure, e.g., out of convenience for a small minimal sample size. Hartmeyer and Andersen [19] show very comprehensively that the various metrics have equally various theoretical considerations and are thus appropriate for research on different aspects of light exposure and related outcomes. Metrics should be chosen a priori to the power analysis and should depend on the wavelength range of interest. As an example, mean first timing of light above threshold would fit analysis of forward circadian phase shifts, whereas interdaily stability (IS) would better fit research questions regarding circadian disruption. The measurement data from wearable devices provide neither of these metrics, but a time series of measurements. Additional software is needed to calculate metrics before applying the described method for power analysis, such as the R package LightLogR [31,32], which provides a coherent interface for all metrics.

## Conclusion

In summary, the emerging and fast-growing field of research utilizing wearable light exposure loggers and optical radiation dosimeters lacks appropriate ways to determine sample sizes based on statistical power. Standard methods do not address the complexity and nested structure inherent in time-series data. Here, we adapted and implemented a procedure based on robust, stepwise bootstrapping to calculate power and required sample size for these data and derived metrics that take the hierarchical data structure into account.

## Supporting information

**S1 File. Quarto HTML document containing code and results for the analysis.**
(HTML)

**S1 Fig. Illuminance values across the dataset.**
(PNG)

**S2 Fig. Melanopic equivalent daylight (mel EDI) illuminance values across day-shift days.**
(PNG)

**S3 Fig. Melanopic equivalent daylight (mel EDI) illuminance values across day-shift days with colored indications of wear and non-wear times.**
(PNG)

**S4 Fig. Ridgeline density plots of melanopic equivalent daylight (mel EDI) between winter and summer across participants.** A red dashed line indicates a threshold of 250 lux.
(PNG)

## Author Contributions

**Conceptualization:** Johannes Zauner, Ljiljana Udovicic, Manuel Spitschan.

**Data curation:** Johannes Zauner, Ljiljana Udovicic.

**Formal analysis:** Johannes Zauner.

**Funding acquisition:** Manuel Spitschan.

**Methodology:** Johannes Zauner, Manuel Spitschan.

**Project administration:** Johannes Zauner, Manuel Spitschan.

**Software:** Johannes Zauner.

**Visualization:** Johannes Zauner.

**Writing – original draft:** Johannes Zauner.

**Writing – review & editing:** Johannes Zauner, Ljiljana Udovicic, Manuel Spitschan.

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
