## [Decision Letter · Decision Letter 0]

15 Sep 2024

PONE-D-24-30405Power analysis for personal light exposure measurements and interventionsPLOS ONE

Dear Dr. Zauner,

Thank you for submitting your manuscript to PLOS ONE. After careful consideration, we feel that it has merit but does not fully meet PLOS ONE’s publication criteria as it currently stands. Therefore, we invite you to submit a revised version of the manuscript that addresses the points raised during the review process.

**ACADEMIC EDITOR: **

**Please follow the recommendation of reviewers and carefully answer their questions.**

Please submit your revised manuscript before Oct 30 2024 11:59PM. If you will need more time than this to complete your revisions, please reply to this message or contact the journal office at plosone@plos.org. Please include the following items when submitting your revised manuscript:A rebuttal letter that responds to each point raised by the academic editor and reviewer(s). You should upload this letter as a separate file labeled 'Response to Reviewers'.A marked-up copy of your manuscript that highlights changes made to the original version. You should upload this as a separate file labeled 'Revised Manuscript with Track Changes'.An unmarked version of your revised paper without tracked changes. You should upload this as a separate file labeled 'Manuscript'.

We look forward to receiving your revised manuscript.

Kind regards,

Ayman A. Swelum

Academic Editor

PLOS ONE

“The project (22NRM05 MeLiDos) has received funding from the European Partnership on Metrology, co-financed by the European Union’s Horizon Europe Research and Innovation Programme and by the Participating States. All authors are part of the project. The project directly funds the position of J.Z.”

Additional Editor Comments:

Please improve clarity, provide more detail on the methodology, expand on the implications of the findings, and provide additional context and examples.

Reviewers' comments:

Reviewer's Responses to Questions

**Comments to the Author**

1. Is the manuscript technically sound, and do the data support the conclusions?

Reviewer #1: Yes

Reviewer #2: Yes

Reviewer #3: Yes

2. Has the statistical analysis been performed appropriately and rigorously? 

Reviewer #1: Yes

Reviewer #2: Yes

Reviewer #3: Yes

3. Have the authors made all data underlying the findings in their manuscript fully available?

Reviewer #1: Yes

Reviewer #2: Yes

Reviewer #3: Yes

4. Is the manuscript presented in an intelligible fashion and written in standard English?

Reviewer #1: Yes

Reviewer #2: Yes

Reviewer #3: Yes

5. Review Comments to the Author

Reviewer #1: In this manuscript by Zauner et al., the authors present a method to estimate the required sample size of study participants to detect statistically significant differences in light exposure metrics with a power of at least 80%. I think that such methods, grounded in statistical robustness, are necessary and timely for the field, owing to the rapid and large expanse of data collecting technologies and methodological advances. This would be a good contribution to the field. I have a few minor concerns which are stated below:

1. The authors highlight that recently there has been a shift to collecting light exposure timeseries data instead of single timepoint spot measurements. This is also associated with the data being considerably messier with several usable and unusable time frames. Conceptually, this situation also appears to be applicable to animal studies. Long term behavioral recordings or luciferase traces are messy (non-stationary) and sometimes have missing data points. However, we use summary statistics of the timeseries such as period and power to characterize the timeseries and to compare between genotypes or manipulations. I wonder if similar techniques, described by the authors in this manuscript, can be a useful way to analyze records from model systems. I think some discussion on this issue may be a good addition to the manuscript and will lead model system researchers to think about this problem.

2. Moreover, when one estimates several metrics from the same timeseries (in this case light exposure), how can one be sure which of these metrics are biologically relevant. I think it is worthwhile for the authors to spend some time discussing that. Further, if different metrics require different sample sizes to reach the desired power for statistical significance between groups, how can one be careful about choosing the metric of relevance and not a metric out of convenience? Cautioning researchers against that would be important.

3. The figures in the manuscript (of the PDF version) are very poor resolution. It would be helpful to have them be hi-res.

Reviewer #2: This manuscript addresses an important and critical issue in the burgeoning field of possible light exposure effects on human health and disease, varying from effects such as SAD (seasonal affective disorder), or effects on the circadian clock, or to other non-visual effects on human physiology such as recent evidence of reduced propagation of the COVID virus in northern latitudes. This paper adresses the problem of achieving standardized, accurate, comparable data on light exposure parameters in a given population that enable comparison and reproducibility between studies in the literature. The manuscript provides a thorough, thoughtful analysis of the parameters, sample size, and statistical treatment needed to extract useful data for a variety of light parameters including exposure time, total power of exposure, and so on. This is a marked improvement on how current decisions on sample size for light exposure data collection are achieved, which lack a specific basis rooted in power analysis and/or are unrelated to power, are possibly completely unclear as to which parameters are important, or have been derived from unrelated fields and therefore never verified in the chosen population. The methods described here are specific to the light exposure metric and the effect size inherent in the light exposure data at the basis of the bootstrap. The method goes beyond typical implementations of bootstrapping to appropriately address the structure of the data and can be readily applied to other datasets that allow comparisons of scenarios beyond seasonal

differences and for different activity patterns. For all these reasons the methods described in the paper represent a real improvement over current methods and will help standardise and promote this important field of public health to advance and provide credible data.

A minor comment is that the authors do not provide examples or discuss studies in which effects of light on humans have been addressed in any detail, which would surely add interest to the paper for the general reader. For example, in the introduction,these studies on light effects on humans are just cited in passing; it would be helpful if they were more fully described in the introduction and in the conclusion the authors provided a few examples on how results currently published in the literature could have benefitted from analysis by their method. A related issue is to provide examples of cases where light intensity (power) for example is the most relevant parameter affecting human response, and others where total exposure time for instance is the most relevant parameter. In other ways, be more specific on the studies that could be better addressed and how to use the methodology in these studies. These are not criticisms of the paper, but would be helpful for people in the field who may think of using the methodology. Another comment along these lines is wavelength. It is well known that UV has far different effects from longer wavelength light on human health, could their method be adapted to detect wavelength exposure. Finally the manuscript is clear and understandable, but there are a lot of 'Germanisms' in the phrases and expressions which, though not technically wrong, interrupt the flow and make the text more complicated to understand. Perhaps the authors could have a native english speaker edit the final version.

Reviewer #3: Reviewer Report

Manuscript Title:

“Power analysis for personal light exposure measurements and interventions”

1. Summary of the Manuscript

The manuscript addresses the challenge of estimating sample size by introducing a hierarchical bootstrapping procedure. This approach effectively handles the complex hierarchical structure of the light exposure data through stepwise resampling and mixed-effects model. The proposed method is applied and validated in twelve summary metrics of light exposure, revealing a significant variability in sample size requirements among different metrics, with some metrics requiring substantially larger sample sizes than others, highlighting the importance of context-specific considerations in experimental design.

2. Recommendation

The manuscript presents a valuable and novel method for sample size estimation in light exposure studies. I recommend major revisions to improve clarity, provide more detail on the methodology, and expand on the implications of the findings. Providing additional context and examples will enhance the overall clarity and impact of the manuscript.

3. General Comments

The manuscript contributes to the field by offering a novel approach for sample size estimation in studies beyond seasonal differences and for different activity patterns. The introduction of hierarchical bootstrapping is a valuable advancement given the complexity of light exposure data. The dataset provided further enhances the manuscript's practical utility. However, additional details and context would strengthen the overall presentation. The manuscript would also benefit from more information on the validation of the method or comparisons with other approaches to fully demonstrate its effectiveness.

4. Specific Comments

1) The abstract and introduction set the context well but could more detailed explain why current sample size decisions are inadequate or unrelated to power analysis. More comprehensive comparisons between the proposed method and the existing method would underscore the significance of the new method. For example, the manuscript contrasts the proposed bootstrapping method with the Kumle et al. approach. It would be helpful to provide a quantitive comparison of the strengths and limitations of both methods, particularly in terms of their applicability to different types of datasets or research questions.

2) The authors emphasize using their method on the time series dataset, a clearer explanation of its key advantages over traditional methods on time series data would provide clearer insight into its benefits.

3) The manuscript uses the dataset from early-day shift workers to validate the proposed method, but the limitations of this dataset should be addressed more explicitly.

5. Minor Issues

1) The figure captions need to be improved.

2) Consider changing "wearable light loggers" to "wearable light exposure loggers" for clarity.

3) The phrase “a large range of required sample sizes from 3 to >50” could be rephrased to “sample sizes ranging from as few as 3 to more than 50” for better readability.

4) In the power analysis section, the meaning of coefficient beta_i needs to be clarified.

5) The method section needs to be more organized.

6. PLOS authors have the option to publish the peer review history of their article (what does this mean?). If published, this will include your full peer review and any attached files.

Reviewer #1: **Yes: **Lakshman Abhilash

Reviewer #2: No

Reviewer #3: No

---

## [Author Response · Author response to Decision Letter 0]

28 Oct 2024

We thank the editor and the reviewers for their insights and feedback. We have made several changes to the manuscript based on this feedback. We reference these changes point by point, together with the reviewer’s remarks. Line number refer to the revised manuscript without track changes.

Reply to Journal requirements:

Our reply: We changed the manuscript according to the templates

“The project (22NRM05 MeLiDos) has received funding from the European Partnership on Metrology, co-financed by the European Union’s Horizon Europe Research and Innovation Programme and by the Participating States. All authors are part of the project. The project directly funds the position of J.Z.”

Our reply: We have updated our financial statements in the the cover letter according to your requirement

Our reply: We would like to note that the data is already being shared and freely accessible through the provided links. What we will do after acceptance is to make an archive link with a persistent identifier. We refrain from doing that now, as a reviewer might require changes to the analysis code that is part of the data repository. Also the code check for reproducibility can only be done after acceptance, as mentioned in the manuscript.

Reply to Reviewers

Reviewer #1: In this manuscript by Zauner et al., the authors present a method to estimate the required sample size of study participants to detect statistically significant differences in light exposure metrics with a power of at least 80%. I think that such methods, grounded in statistical robustness, are necessary and timely for the field, owing to the rapid and large expanse of data collecting technologies and methodological advances. This would be a good contribution to the field. I have a few minor concerns which are stated below:

Our reply: We thank the reviewer for the encouragement about the publication and the specific feedback.

[R1.1] The authors highlight that recently there has been a shift to collecting light exposure timeseries data instead of single timepoint spot measurements. This is also associated with the data being considerably messier with several usable and unusable time frames. Conceptually, this situation also appears to be applicable to animal studies. Long term behavioral recordings or luciferase traces are messy (non-stationary) and sometimes have missing data points. However, we use summary statistics of the timeseries such as period and power to characterize the timeseries and to compare between genotypes or manipulations. I wonder if similar techniques, described by the authors in this manuscript, can be a useful way to analyze records from model systems. I think some discussion on this issue may be a good addition to the manuscript and will lead model system researchers to think about this problem.

Our reply: We have added a sentence to the discussion to open the method beyond light exposure (lines 302ff)

[R1.2] Moreover, when one estimates several metrics from the same timeseries (in this case light exposure), how can one be sure which of these metrics are biologically relevant. I think it is worthwhile for the authors to spend some time discussing that. Further, if different metrics require different sample sizes to reach the desired power for statistical significance between groups, how can one be careful about choosing the metric of relevance and not a metric out of convenience? Cautioning researchers against that would be important.

Our reply: We agree and have added a section to the discussion/limitations to address this concern (lines 324-334)

[R1.3] The figures in the manuscript (of the PDF version) are very poor resolution. It would be helpful to have them be hi-res.

Our reply: The figures we upload are hi-res. As far as we know, PLOS only includes low-res versions in PDFs, but a download link to the full-res versions should be on the top right of each image page and should say sth. similar to “Click here to access/download;Figure;Figure1.tif”

Reviewer #2: This manuscript addresses an important and critical issue in the burgeoning field of possible light exposure effects on human health and disease, varying from effects such as SAD (seasonal affective disorder), or effects on the circadian clock, or to other non-visual effects on human physiology such as recent evidence of reduced propagation of the COVID virus in northern latitudes. This paper adresses the problem of achieving standardized, accurate, comparable data on light exposure parameters in a given population that enable comparison and reproducibility between studies in the literature. The manuscript provides a thorough, thoughtful analysis of the parameters, sample size, and statistical treatment needed to extract useful data for a variety of light parameters including exposure time, total power of exposure, and so on. This is a marked improvement on how current decisions on sample size for light exposure data collection are achieved, which lack a specific basis rooted in power analysis and/or are unrelated to power, are possibly completely unclear as to which parameters are important, or have been derived from unrelated fields and therefore never verified in the chosen population. The methods described here are specific to the light exposure metric and the effect size inherent in the light exposure data at the basis of the bootstrap. The method goes beyond typical implementations of bootstrapping to appropriately address the structure of the data and can be readily applied to other datasets that allow comparisons of scenarios beyond seasonal

differences and for different activity patterns. For all these reasons the methods described in the paper represent a real improvement over current methods and will help standardise and promote this important field of public health to advance and provide credible data.

Our reply: We thank the reviewer for the encouragement about the method and the specific feedback.

[R2.1] A minor comment is that the authors do not provide examples or discuss studies in which effects of light on humans have been addressed in any detail, which would surely add interest to the paper for the general reader. For example, in the introduction, these studies on light effects on humans are just cited in passing; it would be helpful if they were more fully described in the introduction and in the conclusion the authors provided a few examples on how results currently published in the literature could have benefitted from analysis by their method.

Our reply: We have added a section to the introduction that exemplifies the dependency from some of the studies (lines 68ff)

[R2.2] A related issue is to provide examples of cases where light intensity (power) for example is the most relevant parameter affecting human response, and others where total exposure time for instance is the most relevant parameter. In other ways, be more specific on the studies that could be better addressed and how to use the methodology in these studies. These are not criticisms of the paper, but would be helpful for people in the field who may think of using the methodology.

Our reply: We agree. The exact biological quantities are not well known, however, such as integration constants under real-world conditions. We have added a section to address your input in the introduction (lines 77ff) and the discussion (lines 324ff).

[R2.3] Another comment along these lines is wavelength. It is well known that UV has far different effects from longer wavelength light on human health, could their method be adapted to detect wavelength exposure. 

Our reply: We fully agree and have expanded our framing to include optical radiation exposure in general. These changes can be found throughout the abstract, introduction, discussion, and conclusion sections.

[R2.4] Finally the manuscript is clear and understandable, but there are a lot of 'Germanisms' in the phrases and expressions which, though not technically wrong, interrupt the flow and make the text more complicated to understand. Perhaps the authors could have a native english speaker edit the final version.

Our reply: We have made several passes over the document to reduce such phrases and expressions. 

Reviewer 3

Summary of the Manuscript

The manuscript addresses the challenge of estimating sample size by introducing a hierarchical bootstrapping procedure. This approach effectively handles the complex hierarchical structure of the light exposure data through stepwise resampling and mixed-effects model. The proposed method is applied and validated in twelve summary metrics of light exposure, revealing a significant variability in sample size requirements among different metrics, with some metrics requiring substantially larger sample sizes than others, highlighting the importance of context-specific considerations in experimental design.

Recommendation

The manuscript presents a valuable and novel method for sample size estimation in light exposure studies. I recommend major revisions to improve clarity, provide more detail on the methodology, and expand on the implications of the findings. Providing additional context and examples will enhance the overall clarity and impact of the manuscript.

Our reply: We thank the reviewer for the encouragement about the method and the specific feedback.

[R3.1] General Comments

The manuscript contributes to the field by offering a novel approach for sample size estimation in studies beyond seasonal differences and for different activity patterns. The introduction of hierarchical bootstrapping is a valuable advancement given the complexity of light exposure data. The dataset provided further enhances the manuscript's practical utility. However, additional details and context would strengthen the overall presentation. The manuscript would also benefit from more information on the validation of the method or comparisons with other approaches to fully demonstrate its effectiveness.

Our reply: We have added more context by providing examples for studies that use these type of data, considerations on the choice of metric, and comparisons with existing methods. As these points are specified in the specific comments, we will provide more details on changes in the manuscript there.

Specific Comments

[R3.2] The abstract and introduction set the context well but could more detailed explain why current sample size decisions are inadequate or unrelated to power analysis. More comprehensive comparisons between the proposed method and the existing method would underscore the significance of the new method. For example, the manuscript contrasts the proposed bootstrapping method with the Kumle et al. approach. It would be helpful to provide a quantitive comparison of the strengths and limitations of both methods, particularly in terms of their applicability to different types of datasets or research questions.

Our reply: We have expanded the discussion section to address this point (lines 288ff).

[R3.3] The authors emphasize using their method on the time series dataset, a clearer explanation of its key advantages over traditional methods on time series data would provide clearer insight into its benefits.

Our reply: As the time series data are condensed into metrics, the power analysis is not directly using the time series itself. We have added a section to the discussion section in lines 330 through 334 to make this more clear.

[R3.4] The manuscript uses the dataset from early-day shift workers to validate the proposed method, but the limitations of this dataset should be addressed more explicitly.

Our reply: The paragraph from line 309 through 317 is dedicated to the limitations of the dataset on the generalizability of our results section.

Minor Issues

[R3.5] The figure captions need to be improved.

Our reply: We expanded all figure captions

[R3.6] Consider changing "wearable light loggers" to "wearable light exposure loggers" for clarity.

Our reply: We changed all instances

[R3.7] The phrase “a large range of required sample sizes from 3 to >50” could be rephrased to “sample sizes ranging from as few as 3 to more than 50” for better readability.

Our reply: We changed the phrasing

[R3.8] In the power analysis section, the meaning of coefficient beta_i needs to be clarified.

Our reply: The coefficient is clarified in lines (231 and 232)

[R3.9] The method section needs to be more organized.

Our reply: We reorganized the method section and added sub-headings for longer portions

---

## [Editor Report · Decision Letter 1]

30 Oct 2024

Power analysis for personal light exposure measurements and interventions

PONE-D-24-30405R1

Dear Dr. Johannes Zauner,

We’re pleased to inform you that your manuscript has been judged scientifically suitable for publication and will be formally accepted for publication once it meets all outstanding technical requirements.

Kind regards,

Ayman A. Swelum

Academic Editor

PLOS ONE

---

## [Editor Report · Acceptance letter]

2 Dec 2024

PONE-D-24-30405R1 

PLOS ONE

Dear Dr. Zauner, 

I'm pleased to inform you that your manuscript has been deemed suitable for publication in PLOS ONE. Congratulations! Your manuscript is now being handed over to our production team.

Kind regards, 

on behalf of

Professor Ayman A. Swelum 

Academic Editor

PLOS ONE